# Deuterium-depletion has no significant impact on the mutation rate of *Escherichia coli*, deuterium abundance therefore has a probabilistic, not deterministic effect on spontaneous mutagenesis

Walliyulahi Ajibola[1,2☯], Ildikó Karcagi[1☯], Gábor Somlyai[3], Ildikó Somlyai[3], Tamás Fehér[1]*

1 Synthetic and Systems Biology Unit, Institute of Biochemistry, Biological Research Centre of the Eötvös Lóránd Research Network, Szeged, Hungary, 2 Faculty of Science and Informatics, Doctoral School in Biology, University of Szeged, Szeged, Hungary, 3 HYD LLC for Cancer Research and Drug Development, Budapest, Hungary

☯ These authors contributed equally to this work.
* feher.tamas@brc.hu[

## Abstract

Deuterium (D), the second most abundant isotope of hydrogen is present in natural waters at an approximate concentration of 145–155 ppm (ca. 1.5E-4 atom/atom). D is known to influence various biological processes due to its physical and chemical properties, which significantly differ from those of hydrogen. For example, increasing D-concentration to >1000-fold above its natural abundance has been shown to increase the frequency of genetic mutations in several species. An interesting deterministic hypothesis, formulated with the intent of explaining the mechanism of D-mutagenicity is based on the calculation that the theoretical probability of base pairs to comprise two adjacent D-bridges instead of H-bridges is 2.3E-8, which is equal to the mutation rate of certain species. To experimentally challenge this hypothesis, and to infer the mutagenicity of D present at natural concentrations, we investigated the effect of a nearly 100-fold reduction of D concentration on the bacterial mutation rate. Using fluctuation tests, we measured the mutation rate of three *Escherichia coli* genes (*cycA*, *ackA* and *galK*) in media containing D at either <2 ppm or 150 ppm concentrations. Out of 15 pairwise fluctuation analyses, nine indicated a significant decrease, while three marked the significant increase of the mutation/culture value upon D-depletion. Overall, growth in D-depleted minimal medium led to a geometric mean of 0.663-fold (95% confidence interval: 0.483–0.911) change in the mutation rate. This falls nowhere near the expected 10,000-fold reduction, indicating that in our bacterial systems, the effect of D abundance on the formation of point mutations is not deterministic. In addition, the combined results did not display a statistically significant change in the mutation/culture value, the mutation rate or the mutant frequency upon D-depletion. The potential mutagenic effect of D present at natural concentrations on *E. coli* is therefore below the limit of detection using the indicated methods.

**Data Availability Statement:** All relevant data are within the paper and its Supporting Information files.

**Funding:** This work was supported by the National Research, Development, and Innovation Office, Hungary (grant No. OTKA K 119298, awarded to T. F.) (www.nkfih.gov.hu); and by the Economic Development and Innovation Operative Program of the EU (grant No. GINOP-2.3.2-15-2016-00001, awarded to T.F.) (www.ginop.hu). The funders had no role in study design, data collection and analysis, decision to publish, or preparation of the manuscript.

**Competing interests:** GS and IS are employed by HYD LLC. HYD LLC for Cancer Research and Drug Development is involved in drug registration based on deuterium depletion and in the development of deuterium-depleted nutritional products. The above company is not traded publicly nor does it distribute shares or annuities. This does not alter our adherence to PLOS ONE policies on sharing data and materials.

## Introduction

Hydrogen (H or $^1$H) and deuterium (D or $^2$H) have the largest mass ratio among stable isotopes of the same element, resulting in significantly different chemical and physical properties [1–3]. Surface waters contain approx. 16 mmol/L (145–155 ppm) deuterium, mainly in the form of $^1$H$^2$HO. In living organisms, the D concentration is approximately 12 mmol/L, but there are significant differences in intramolecular D distribution [4]. For example, plants are divided into three groups (C3, C4 and CAM) depending on which one of three different pathways they harness to fix carbon dioxide during photosynthesis [5]. CAM-group plants enrich D in carbohydrates, while C3 and C4 plants deplete deuterium [6]. It was also shown that the stable isotope ratio (D/H) of carbon-bound hydrogen at given positions of the six carbon atoms of a glucose molecule within C3 and C4 plants can be higher or lower than expected (i.e. D-enriched or D-depleted, respectively) based on the natural abundance of D [7,8]. Lipids show an even more dramatic D-depletion compared to carbohydrates, further supporting that certain biochemical pathways are highly selective for the various hydrogen isotopes [9]. The presence of D within substrates can strongly influence enzyme activities [10] with the yeast H$^+$-ATPase being the ultimate example of not accepting D as substrate at all [11].

After the discovery of D in 1932, it was shown in numerous experiments that extreme (>1000-fold elevated) D concentrations have major, mainly toxic effects to all kingdoms of life (recently reviewed in reference [12]): bacteriophages [13], bacteria [14], yeast [15], algae [16], plants [17], *Drosophila melanogaster* [18] and mammalian cell lines [19] are, to various extents, impaired in growth or reproduction. When exposing mammals to D$_2$O, a wide range of symptoms, including lymphopenia, hypoglycemia, myocardial abnormality, generalized muscular weakness and impaired reproductive capacity were observed [20,21]. In humans, the effect of such severe D-intoxication has not been studied for ethical reasons, but a 100-fold elevation of cellular D concentration for 6 weeks led to only mild and transient symptoms [22,23].

The mutagenic effect of deuterium has also been in the focus of multiple investigations for several species. The presence of high (50–99.8%) D$_2$O concentration was shown to elevate mutation frequencies in bacteriophage T4 [13], in various bacteria including *Escherichia coli* [14], *Salmonella typhimurium* [24], *Proteus mirabilis* [25] and *Streptococcus pneumoniae* [26], as well as in rice (*Oryza sativa*) [17]. The observable changes in mutation frequencies were often strain and locus-dependent, but on average meant an order of magnitude of increase. Two investigations reported no mutagenic effect of D$_2$O in *D. melanogaster* [27] and *E. coli* [28], respectively, which may have resulted from the low number of parallels used. A study also proved that deuterium-oxide enhanced the mutation rate induced by gamma rays in cultured mouse leukemia cells [29]. More recently, the in vitro error-promoting effect of 99% D$_2$O was demonstrated for Taq DNA polymerase, increasing the frequency of mutations within PCR products approx. nine-fold [30].

Among the numerous explanations regarding the mutagenic effect of D, perhaps the most interesting is the one formulated by Pedersen and co-workers [31]. The basis of their explanation is the fact that the probability of base pairs harboring two adjacent deuterium-bonds (D-bridges) instead of hydrogen-bonds (H-bridges) (i.e. 1.5 x $10^{-4}$ squared) is approximately equal to the mutation rate (~2 x $10^{-8}$ mutations/nucleotide/generation) calculated for humans and *Caenorhabditis elegans* [31]. This model implicitly assumes that the presence of D is the only cause of single nucleotide exchange events, and that practically each such double D-bridge leads to a point mutation. If this deterministic feature was correct, a 2-order of magnitude decrease in the labile D-content of DNA should yield a 4-order of magnitude decrease in the mutation rate, considering that the frequency of double D-bridges is proportional to the square of D-abundance.

Apart from mutagenicity tests, the past decade has brought about an increasing amount of data indicating that the D content of natural waters may have a significant effect on biological processes, e.g. concerning brain function [32], cellular aging [33] and tumor growth [34]. Although the mutagenic effect of $D_2O$ seems to be convincingly demonstrated at >1000-fold increased concentrations, no study has yet investigated the potential effect of D on the rate of mutations at its natural abundance.

In this work, we measure the effect of reducing the D content of the culture medium on the mutation rate of *E. coli* for two reasons: to test the hypothesis of Pedersen et al., and to assess the contribution of natural D concentrations to the spontaneous mutagenesis of bacterial cells. We report a minor decrease in mutation rate upon D-depletion, not meeting most of our criteria for statistical significance. The observed effect stood far from the expected 10,000-fold (= 100 squared) reduction, indicating that the effect of D abundance on mutagenesis is not deterministic.

## Materials and methods

### Chemicals and media

Bacterial cultures were grown in Mineral Salts (MS) medium [35] complemented with either 10.1 mM glucose or 217.3 mM glycerol. To make a D-depleted medium, the components of MS (17.3 mM $K_2HPO_4$, 51.5 mM $KH_2PO_4$, 7.5 mM $(NH_4)_2SO_4$, 1.44 mM Na-citrate, 0.41 mM $MgSO_4$, 2 μM $FeSO_4$) were added in the form of crystals to D-depleted water (≤1 ppm deuterium oxide; Sigma-Aldrich, St. Louis, MO, USA), followed by the carbon source (glucose crystals or 99% glycerol) and filtered sterile using a 0.22 μm pore-size Millex®GP PES filter (Merck-Millipore, Carrigtwohill, Ireland). For the *cycA* tests, sterile vitamin B1 injection (Zentiva, Prague, Czech Republic) was directly added to obtain a 2.5 μg/mL thiamine concentration. Control medium was made from D-depleted medium by re-complementing with $D_2O$ (99.9 atom%; Sigma-Aldrich, St. Louis, MO, USA) to obtain 150 ppm deuterium concentration. MS components were provided by Molar Chemicals Ltd. (Budapest, Hungary). Mutant selection was carried out on 1.5% agar-MS plates, prepared using water of 150 ppm D. The plates contained glucose as the carbon source (glycerol for *ackA* and *galK* tests), and one of the following selective agents: cycloserine (0.04 mM), chloroacetate (10 mM) or galactose (10 mM) (all obtained from Sigma-Aldrich, St. Louis, MO, USA).

### Analysis of bacterial growth

Bacterial growth was analyzed by recording the growth curves in MS + 0.2% glucose media containing <2 or 150 ppm D. Ten parallel bacterial cultures of 100 μL for each condition were shaken in a Synergy 2 microplate reader (BioTek, Winooski, VT, USA) at 37 °C. The optical density (OD) was recorded at 600 nm at 15 minute-intervals for 24 h, and plotted against time to obtain the growth curves.

### Measurement of D-content

The D-content of the culture media described above was verified using a Liquid-Water Isotope Analyser-24d manufactured by Los Gatos Research Inc. (Mountain View, CA, USA). The instrument uses off-axis integrated cavity output spectroscopy to measure absolute abundances of $^2HHO$ via laser absorption [36]. Isotopic ratios are reported as δD value with respect to the international VSMOW standard in permill according to the following formula:

$$\delta D = \frac{R_{sample} - R_{standard}}{R_{standard}} * 1000 [‰]$$

where $R_{sample}$ and $R_{standard}$ represent the $^2H/^1H$ (D/H) ratios in the sample and the standard, respectively. The uncertainty of δD was ±3.0 [‰]$_{VSMOW}$. The measured deuterium concentrations were converted to ppm (with ±1 ppm uncertainties).

Since this instrument was validated to measure between 15–150 ppm, D-depleted solutions were measured after being mixed with a solution of high D-content (120–150 ppm, pre-measured) in a pre-defined ratio, allowing the calculation of the D-content of the original depleted solution.

## Bacterial strain engineering

The *E. coli* K-12 MG1655 strain was used in the experiments detecting the mutations of *cycA* and *ackA*, and its derivate strain *E. coli ΔgalU* was used to infer the mutation rate of the *galK* gene. To generate the *E. coli ΔgalU* strain, the *galU* gene (GenBank GeneID: 945730) was deleted using linear-DNA mediated recombineering [37]. In brief, the Streptomycin resistance gene (SmR) was PCR-amplified from plasmid pCDM4 [38] using primers galU_SpF and galU_SpR, which carry homology arms specific for *galU* (S1 Table). The PCR product was electroporated into *E. coli* MG1655 carrying the arabinose-induced pKD46 plasmid (GenBank ID: AY048746.1). Recombinants were selected on LB+Sp plates. Colonies were screened by colony-PCR using primers galU_D and SmFw. One positive clone was analyzed by Sanger-sequencing using primers galU_D and galU_E to verify the replacement of the *galK* gene by SmR. Plasmid pKD46 was a kind gift of Dr. Barry Wanner, pCDM4 was a kind gift of Dr. Mattheos Koffas.

## Fluctuation analyses

To compare the mutation rates in D-depleted and control media, we applied a fluctuation analysis protocol described earlier [39] on three different mutant selection assays. Spontaneous mutations of the *cycA* (GenBank GeneID: 948725), *ackA* (ID: 946775) and *galK* (ID: 945358) genes were selected on cycloserine, chloroacetate and galactose plates, respectively. In each assay, two overnight starter cultures of *E. coli* were fully grown in control and D-depleted MS medium + carbon source (glucose for *cycA*, glycerol for *ackA* and *galK* assays), and diluted in the respective fresh media to obtain densities of $10^5$ cells/mL. The fresh cultures were each divided into 20 parallel wells (10 parallels for *cycA* assay) of a 96-well microplate (Greiner Bio-One International, Kremsmünster, Austria) to generate parallel cultures of 100 μL, and shaken at 37˚C in a Synergy 2 microplate reader/incubator (BioTek, Winooski, VT, USA) until the point of entering the stationary phase (usually 20–24 h for glucose, 36 h for glycerol). Considering that the final densities were $10^9$ cells/mL, this growth corresponded to ca. 13 cell duplications or generations. Optical densities (OD) of the cultures were monitored throughout the incubation at 600 nm. Next, fractions of each culture were plated on selective plates (50 μL on cycloserine-, 5 μL on chloroacetate- and 25 μL on galactose-containing plates), and incubated at 37 °C for 30 h (48 h for *ackA* and *galK* assays). Appropriate dilutions of three parallel cultures from each condition were also plated on MS + glucose medium (MS + glycerol for *ackA* and *galK* tests) to obtain mean total cell counts (*N*) for both control and D-depleted media. The colonies counted on each selective plate were used in our Excel-based fluctuation analysis calculator to calculate *m*, the number of mutations/culture for each medium. We applied the Ma-Sandri-Sarkar Maximum Likelihood (MSS-MLL) approach, claimed to be the most robust method of fluctuation analysis valid in all investigated ranges of mutation rates [40]. Since only a fraction of each culture was plated, the $m_{actual}$ value was extrapolated from *m* using the formula $m_{actual} = m(z-1)/(z^*\ln(z))$, where z indicates the plated fraction (0.5 for *cycA*, 0.05 for *ackA* and 0.25 for *galK* assays) [40]. Mutation rates (μ) were calculated for each medium using

the μ = $m_{actual}/N$ formula to give the number of mutations/gene/generation. Statistical methods used for pair-wise comparison and combined analysis of the obtained values is described in detail in the Statistical analysis section. The measured values were defined incomparable and the experiment was discarded if (a) the total cell counts displayed a significant difference or if (b) the compared mutation rates (μ displayed a pattern that was opposite to the pattern of the compared $m_{actual}$ values. The calculated values originating from discarded experiments and valid experiments can be found in S2 Table, marked as plain and bold text, respectively. Raw data (colony counts) corresponding to all experiments are available in S1 File.

## Determining mutant frequencies

Mutant frequencies were determined from the experimental data generated in the course of the fluctuation tests described above. For each group of bacterial cultures grown under a specific condition in each experiment, the median number of mutant colonies obtained on selective plates was calculated, and divided by the corresponding mean total cell number ($N$) to obtain the mutant frequencies ($f$). The significance of the change in $f$ caused by elevating the D content from <2 to 150 ppm was tested for each experiment using standard tables generated for this purpose [41]. As for the combined analysis, the median values of the 15 experimental results obtained for both conditions were compared using the Mann-Whitney U test with the online calculator available at www.statskingdom.com.

## Statistical analysis

The $m$ value pairs obtained with each fluctuation analysis were compared pair-wise the following way: since ln($m$), unlike $m$ or $μ$ values, follows a normal distribution, it is the most recommended parameter to use for statistical analysis [40]. We therefore made pair-wise comparisons of the ln($m_{actual}$) values corresponding to <2 ppm and 150 ppm D (referred to as $m_2$ and $m_{150}$, respectively) for each experiment using a two-tailed, unpaired t-test specifically developed for this purpose [40]:

$$t = \frac{\ln(m_{150}) - \ln(m_2)}{\sqrt{\sigma_{150}^2/C_{150} - \sigma_2^2/C_2}}$$

where $\sigma \approx 1.225 m^{-0.315}/\sqrt{C}$, and $C$ is the number of parallel cultures used in an experiment for each condition [40]. The P values corresponding to the calculated $t$ values were obtained from www.vassarstats.net. The degrees of freedom (df) were $C_{150}+C_2-2$. The means of total cell counts ($N$) were compared by applying two-tailed, unpaired t-tests, using Excel. The df and the $t$ values are indicated for all pair-wise comparisons in S2 Table. The ln($m_{actual}$) values were defined incomparable and the experiment was discarded if (a) the total cell counts displayed a significant difference or if (b) the compared mutation rates displayed a pattern that was opposite to the pattern of the compared $m_{actual}$ values. The 95% confidence limits of $m_{actual}$ were calculated for 2 ppm and 150 ppm D for each experiment by raising $e$ (the base of natural logarithm) to the 95% confidence limits of ln($m_{actual}$). The latter were calculated as follows: $CL_{\ln(m)95^+} = \ln(m) + 1.96\sigma(e^{1.96\sigma})^{-0.315}$ and $CL_{\ln(m)95^-} = \ln(m) + 1.96\sigma(e^{1.96\sigma})^{+0.315}$ [40].

The $m_{actual}$ values derived from the 15 valid experiments, corresponding to each D concentration were combined after natural logarithmic transformation to yield their geometric means. In our combined analysis, these were compared by testing for overlapping 95% confidence limits, using the calculations described above. The mean $μ$ values for <2 ppm and 150 ppm D derived from the 15 valid experiments were compared using two-tailed, unpaired t tests.

The threshold of significance (α) was 0.05 for all statistical analyses. Box-plots were displayed using the on-line R application available at http://shiny.chemgrid.org/boxplotr/.

## Results

In this study, our main goal was to compare the mutation rates of bacterial cultures grown in D-depleted and D-containing media, the latter containing D corresponding to its natural concentrations. Since nutritional stresses have been shown to affect the mutational landscape of bacteria [42], special care was taken to limit all differences between the two types of media to their D-abundance. This was achieved by preparing MS medium using water of <1 ppm D content for the investigation, and re-complementing it with pure $D_2O$ to a D-content of 150 ppm for use as the control. The labile $^2H$-content of the salts, glucose and the thiamine solution used to prepare MS are expected to cause a further <1 ppm increase of D concentration for the D-depleted samples, but the resulting <2 ppm D would still be ca. two orders of magnitude lower compared to the control. On one random occasion, the D content of the depleted and re-complemented MS media were measured, and found to be <2 ppm and 161 ppm, respectively, verifying the nearly 100-fold ratio of D-levels. The growth curves of the *E. coli* cultures recorded in the two media were nevertheless indiscernible, indicating identical nutritional conditions and the lack of measurable toxicity of D or D-depletion (Fig 1).

To infer the mutation rates, we relied primarily on a system developed earlier in our lab, which has been successfully used to compare mutation rates of various *E. coli* strains [43–47]. This method detects mutations inactivating the *cycA* gene of *E. coli* (open reading frame (ORF): 1413 bp), which results in the inability of cycloserine uptake and a consequential resistance to this antibiotic. We used the most robust fluctuation analysis protocol (MSS-MLL) available to derive the number of mutations/culture ($m_{actual}$) from the mutant numbers observed in mutant selection assays [40].

Out of the 14 *cycA* assays made (each relying on 2x10 cultures), seven did not fill the stringent criteria listed in the Methods and Materials, and were discarded. Out of the remaining

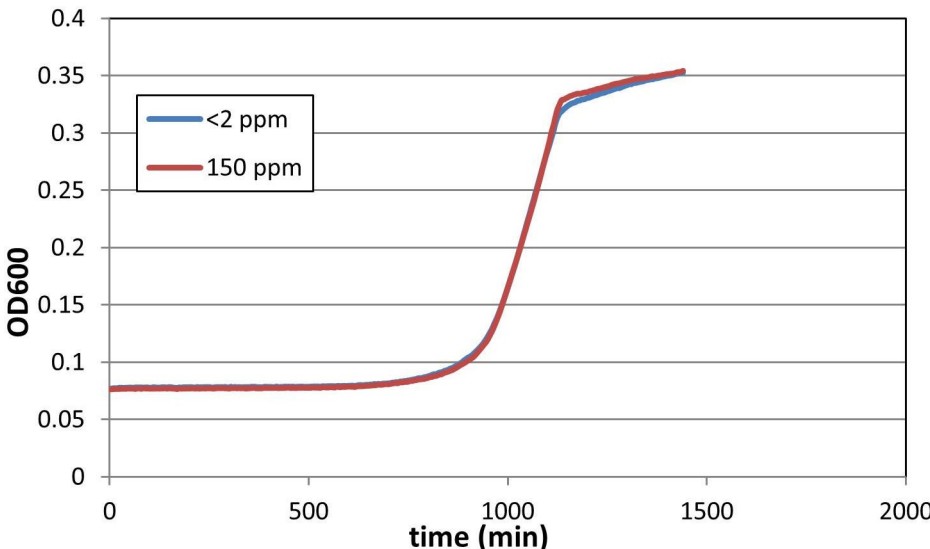

**Fig 1. Effect of D on the growth of *E. coli*.** Cultures were grown in a microplate shaker/incubator in glucose-minimal medium at 37 ºC, and monitored by measuring their absorbance at 600 nm. The growth medium contained either <2 ppm (blue line) or 150 ppm (red line) D. The averages of ten parallel cultures for each condition are shown.

seven valid experiments, D-depletion significantly reduced the $m_{actual}$ value in five cases, with the effect ranging from an 8.3% to a 38.6% decrease (taking $m_{150}$ as 100%) (Fig 2A, S2 Table). Among the two remaining assays, a 23.2% increase was detected upon D-depletion in one, and no significant difference was detectable in the other. The geometric mean of $m_{actual}$ values measured in depleted and normal D concentrations were 17.02 (95% confidence interval (CI95): 13.65–20.66) and 21.72 (CI95: 17.53–26.25) mutations/culture, respectively ($m_2/m_{150} = $ 78.4%). The overlapping confidence limits indicate that we cannot declare with 95% confidence that the two $m_{actual}$ values actually differ.

Our next mutant selection assay selected for mutations inactivating the *ackA* gene (ORF: 1203 bp). Such cells lose acetate kinase activity, therefore do not phosphorylate chloroacetate and thus avoid chloroacetyl phosphate toxicity [48]. We made five *ackA* assays (each relying on 2x20 cultures), and discarded one result for not meeting the minimal criteria. Out of the four valid results, two showed mutation-reduction, one showed mutation induction, and one showed no significant change of the $m_{actual}$ value upon D-depletion (Fig 2B, S2 Table). The geometric mean of $m_{actual}$ values measured in depleted and normal D concentrations were 32.92 (CI95: 27.18–39.06) and 39.54 (CI95: 32.75–46.79) mutations/culture, respectively ($m_2/m_{150} = $ 83.3%). Again, the overlapping confidence limits indicated that we cannot declare with 95% confidence that the two $m_{actual}$ values actually differ.

Finally, we generated Δ*galU* mutants of *E. coli* MG1655 in order to apply the *galK* assay. *E. coli* cells phosphorylate galactose by galactokinase, encoded by *galK*. Our engineered Δ*galU* cell line however lacks the UTP—glucose-1-phosphate uridyltransferase enzyme, and is therefore incapable of channeling galactose-1-phosphate (galactose-1-P) into the cellular metabolism. This strain therefore accumulates galactose-1-P when grown in the presence of galactose, a lethal phenotype rescued by mutations inactivating the *galK* gene (ORF: 1149 bp). We carried out five fluctuation tests on mutants selected on galactose plates, each utilizing 2x20 cultures. Four assays were valid, two of which indicated the decrease, one displayed the increase, and one showed no significant change of $m_{actual}$ upon D-depletion (Fig 2C, S2 Table). The geometric mean of $m_{actual}$ values measured in depleted and normal D concentrations were 34.25 (CI95: 28.30–40.61) and 36.56 (CI95: 30.24–43.31) mutations/culture, respectively ($m_2/m_{150} = $ 93.7%). Again, we could not exclude with 95% confidence the inequality of the two $m_{actual}$ values due to their highly overlapping confidence limits.

The data displayed up to this point dealt with mutation/culture values for the sake of statistically valid comparisons. However, the mutation rate value ($\mu$) is often considered a more

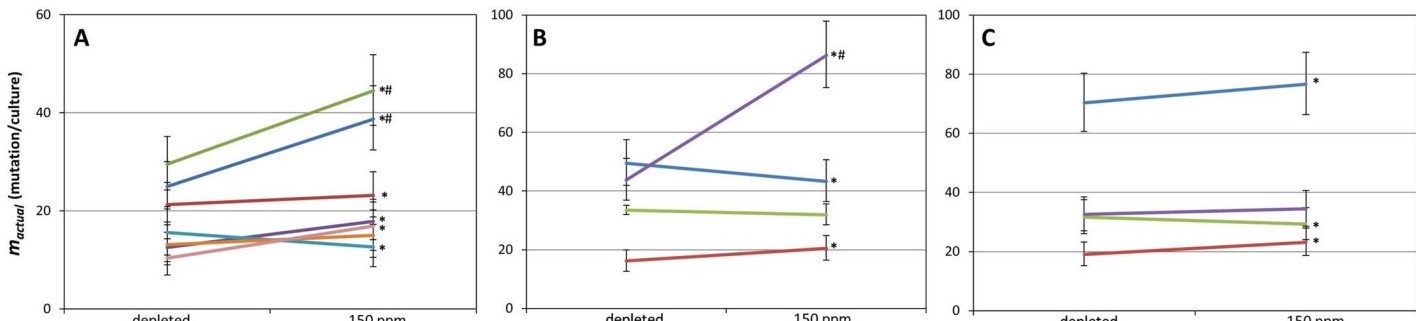

**Fig 2. Effect of D on the number of mutations occurring in *E. coli* cultures.** The results of multiple pairwise fluctuation analyses are shown, made in growth media containing <2 ppm or 150 ppm D using three different mutation detection systems, as described in the Materials and Methods. Only those results are shown which passed our filtering. Asterisks mark the significant (P < .05) difference of the two connected $m_{actual}$ values obtained with a two-tailed, unpaired t-test. Error bars depict the 95% confidence limits (CL) of the calculated $m_{actual}$ values. # marks cases where the 95% CL of a pair do not overlap (**A**) Results obtained using the *cycA* system. (**B**) Results obtained using the *ackA* system. (**C**) Results obtained using the *galK* system.

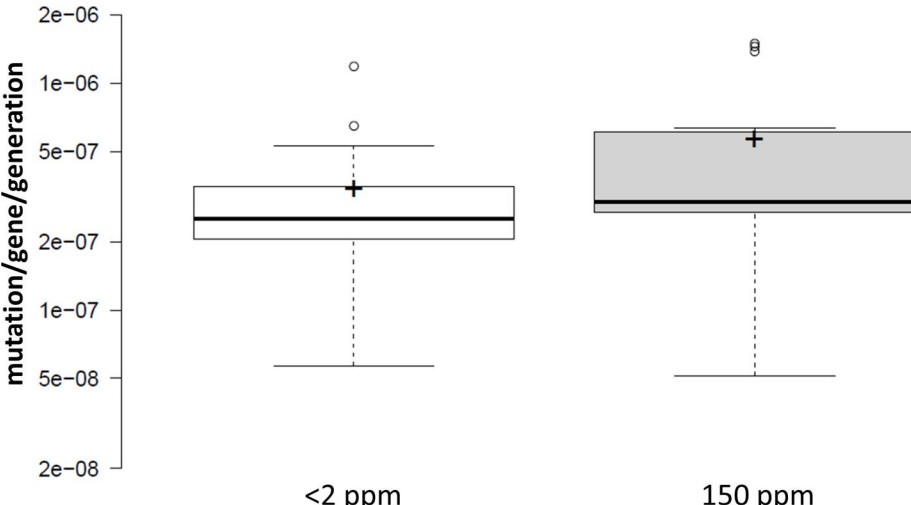

**Fig 3. Combined effect of D on the mutation rates measured in one of three genes (*cycA*, *ackA* or *galK*) of *E. coli*.**
Center lines show the medians; box limits indicate the 25th and 75th percentiles as determined by R software; whiskers
extend 1.5 times the interquartile range from the 25th and 75th percentiles, outliers are represented by dots; crosses
represent sample means. n = 15 sample points. The means are not significantly different ($t$ = 1.46, df = 28, P = .16,
applying a two-tailed, unpaired t-test).

universal parameter since it allows the coarse comparison of different experimental setups.
Therefore, mutation rates were also calculated for each experiment, and are displayed in Fig 3.
In each case, the effect of D-depletion on mutation rates shows a similar pattern to that of
mutation/culture values, since those that did not were excluded in our filtering explained in
the Methods and Materials (S1 Fig). The mutation rates, which are practically mutation/cul-
ture values normalized by the total cell numbers display a similar or larger decrease in response
to D-depletion using the three experimental systems (the geometric mean $\mu_2/\mu_{150}$ ratio was
50.4%, 86.5% and 81.9% for *cycA*, *ackA* and *galK* assays, respectively), but unpaired t-tests
made for each setup yielded statistically insignificant results (*cycA* assay: $t$ = 1.69, df = 12, P =
.12,; *ackA* assay: $t$ = 0.58, df = 6, P = .58,; *galK* assay: $t$ = 0.26, df = 6, P = .80,). The combined
geometric mean $\mu_2/\mu_{150}$ ratio was 0.663 (CI95: 0.483–0.911), however, an unpaired t-test indi-
cates no significant difference among the combined mutation rates ($t$ = 1.46, df = 28, P = .16)
(Fig 3).

To add a parameter that allows further comparison with published datasets, the mutant fre-
quencies were also calculated for the fluctuation tests that were declared valid. The *f* values cal-
culated for each experiment are shown in S3 Table. Five of the 15 tests found that elevated D
significantly increased *f*, four of which belong to the *cycA* experimental setup. The remaining
ten experiments detected no significant difference in mutant frequencies. The geometric mean
mutant frequency ratios ($f_2/f_{150}$) were 0.57, 0.72 and 0.92 for the *cycA*, *ackA* and *galK* assays,
respectively.

## Discussion

Several theories explaining the mutagenicity of D can be found in the literature, four of which
we list here. The central mechanistic explanation for point mutations, the tautomerization
hypothesis originates from one of the milestone-papers of Watson and Crick [49], who
pointed out that the presence of certain nucleobase-tautomers can alter base-pairing within
the double-stranded DNA. Later, Löwdin [50] explicitly described the rare imino and enol

tautomers of the nucleobases, their faulty base-pairing and the resulting point mutations. He hypothesized that if proton tunneling is required for the formation of the rare tautomers, exchanging H to D would reduce the mutation rate by several orders of magnitude. No such anti-mutagenic effect of D could ever be demonstrated, indicating that the presumed mechanism is incorrect. The tautomerization hypothesis itself, however, was not discredited: multiple pieces of experimental evidence emerged supporting the existence of rare tautomers within DNA, the specific details of their formation however, still await clarification (see references in Brovarets and Hovorun [51]). The effect of D on the dynamics of tautomer-formation has been demonstrated [52], providing a potential mechanism for its mutagenic activity.

The second possible mechanism whereby D can influence the emergence of mutations is *via* the solvent effect: the physiological uncoiling of DNA may be affected by the presence of D in the surrounding solvent. In a cornerstone study, the increased energy demand of DNA denaturation in 99.7% $D_2O$, compared to that in natural water was successfully demonstrated using both spectroscopic and calorimetric methods [53]. This phenomenon may influence the efficiency or precision of the DNA polymerase or the repair machinery in a way that could lead to the increase of the mutation rate. The potential solvent effect of D at its natural abundance, however, has not been investigated.

The isotope effect of D on enzyme function exemplified in the introduction [8,11] may provide a third potential mechanism of D-mutagenicity. H to D changes within the DNA or the enzymes responsible for its replication and repair may impair these processes, resulting in a dose-dependent effect on the rate of single nucleotide exchanges. Again, the detailed investigation of this possibility is still awaiting.

And finally, the fourth hypothesis is the one challenged by our experiments. As explained in the introduction, it attributes mutations to the presence of adjacent D-bonds connecting certain base pairs in the DNA. It is based on the equality of two values: the theoretical frequency of base pairs harboring two adjacent D-bridges instead of H-bridges ($1.5 \times 10^{-4}$ squared) with "the error rate of certain eukaryotic DNA polymerase enzymes" [31], implicitly assuming that the presence of D is the only cause of single nucleotide exchange events, and that practically each such double D-bridge leads to a point mutation. (Despite referring to polymerase error rates, that paper actually cites two works that have calculated the *rate of mutations* in humans [54] and *C. elegans* [55] to be ca. $2 \times 10^{-8}$ /site/generation. Those values therefore incorporated the effect of all DNA repair mechanisms, in addition to that of polymerase errors, just like the mutation rates measured in this present work.) The deterministic nature of the above assumptions would suggest that the 2-order of magnitude decrease in the labile D-content of DNA, achieved in our experiments, should yield a 4-order of magnitude decrease in the mutation rate. This was certainly not observed in our work, indicating that the equality of the two frequencies mentioned above is most probably a mere coincidence. In fact, we believe that the presence of a single D-bridge, and not a double D-bridge among the H-bridges of a DNA base-pair may introduce the highest perturbation to DNA structure and therefore the base-pairing process, potentially impairing the fidelity of replication.

It is quite plausible that multiple mechanisms are responsible for the observed mutagenic effect of D and that the above list is incomplete. The effect of each mechanism may be limited to a discrete range of D-concentration, resulting in an overall non-linear or even inconsistent dose effect. These effects make the evaluation of experiments like ours all the more difficult. But for the very reason that Pedersen's hypothesis is an exclusive (i.e. disallowing parallel mechanisms) explanation for mutations, it is unlikely to be valid in its original form. We emphasize however that the molecular mechanism proposed by the same hypothesis may quite possibly contribute to the effect of D on the mutation rate. According to the calculations of the authors, deuteration of nucleobases at the key hydrogens reduces the energy level of

base-paired nucleotides (especially when two or three D-bridges are formed), and thereby affects the kinetics of base-pairing. This change may influence the competition of nucleotides to access the DNA polymerase to such an extent that above certain D levels, it could cause the increase of their misincorporation rate. Our data suggest however that if this theory holds, the effect at natural D abundance is probabilistic, not deterministic. Besides testing a model that explains the mechanism of mutagenicity, we also attempted to use our data to answer the intriguing question whether the D present in the environment has any measurable effect on the mutation rate of bacteria. At an earlier stage of this study, having had three valid fluctuation analyses made with cycloserine selection, we observed three cases of a significant decrease in mutations/culture, giving the impression of D-depletion having a straightforward effect. To avoid relying on limited data, we increased the number of fluctuation tests, as well as applied two further assay types. This way, we observed increases, decreases and sometimes no change in $m_{actual}$ parallel to changes in D concentration (Fig 2). Overall, out of 15 experiments of D-depletion, 11 resulted in the decrease and 4 in the increase of $m_{actual}$ (9 and 3 significant decreases and increases, respectively). D-depletion therefore elicited an antimutagenic effect three times more often than a mutagenic one, but this pattern was not significantly different from random [$\chi^2_{(2, n = 30)}$ = 2.1, P = .35].

When analyzing individual fluctuation test-pairs, a method more informative than t-tests is to seek non-overlapping 95% confidence intervals, for this type of analysis takes the magnitude of the effect into account as well [40,56]. Only three of the 15 valid fluctuation test-pairs yielded non-overlapping CI95 ranges (Fig 2), which means that in 12 cases, we cannot exclude with 95% confidence the equality of the compared $m_{actual}$ values. Combining all $m_{actual}$ values for <2 ppm and 150 ppm D-concentration relying on all three experimental setups gives geometric means of 24.46 (CI95: 21.82–27.19) and 29.28 (CI95: 26.19–32.50) mutations/culture, respectively (Fig 4). The geometric mean $m_2/m_{150}$ ratio of the combined results was 0.835, (CI95: 0.726–0.961). Still, the overlap of the CI95 ranges disallows the exclusion of the equality of the two $m_{actual}$ values with a 95% confidence.

Considering mutational rates, the numbers are slightly different, but the conclusions are the same. Combining all experimental results, D-depletion seems to change the mutation rates 0.663-fold (geometric mean; CI95: 0.483–0.911) (Fig 3). The differences in the mean $\mu$ values (3.31E-7 vs. 5.46E-7 mutations/gene/generation for $\mu_2$ and $\mu_{150}$, respectively) are not significant using an unpaired t-test (P = .16). Importantly, the three different experimental setups yielded highly similar mean mutation rate values at 150 ppm D abundance (5.36E-7, 5.77E-7 and 5.34E-7 mutations/gene/generation for *cycA*, *ackA* and *galK* assays, respectively). However, mean mutation rates measured at <2 ppm D were less similar (2.43E-7, 3.88E-7 and 4.28E-7 mutations/gene/generation for *cycA*, *ackA* and *galK* assays, respectively), explaining the differences in the geometrical mean $\mu_2/\mu_{150}$ ratios displayed in the results. But again, due to the lack of statistical significance of the effects, it is yet inappropriate to start speculating about the differential sensitivity of the three assays to D-depletion.

To make our results comparable to more datasets available in the literature, the mutant frequencies were also calculated for the mutant selection experiments described above. The mutant frequency is a more classical and less accurate parameter describing mutational events for it makes no correction for the potentially early acquisition of mutations (a.k.a. 'jackpots'). Only five of the 15 tests found D to significantly increase mutant frequency, indicating that in the majority of the cases, the inequality of $f_2$ to $f_{150}$ is not supported. Combining all results (Fig 5) gives median values (3.01E-7 and 4.81E-7 mutant/gene/cell for $f_2$ and $f_{150}$, respectively) that are not significantly different using a Mann-Whitney U test ($U$ = 94, z = -0.74, $n_2$ = $n_{150}$ = 15, P = .46). The geometric mean of the $f_2/f_{150}$ ratio is 0.687.

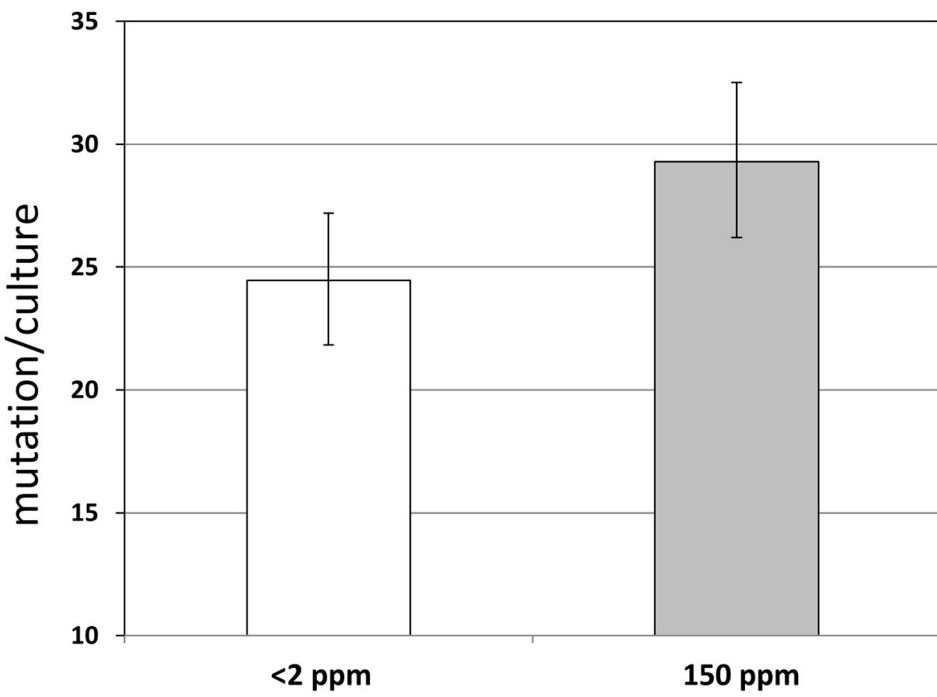

**Fig 4. Combined effect of D on the number of mutations occurring in *E. coli* cultures.** The geometric mean mutation/culture values calculated for D concentrations of <2 ppm (white bar) and 150 ppm (gray bar) are shown, combining the results of all 15 experiments carried out using all three setups. Error bars represent 95% confidence limits.

Accepting that the antimutagenic effect of D-depletion cannot be considered statistically significant based on our data, the question still remains whether the magnitude of the actual changes in mutation rates could possess any biological significance. To address this issue, we

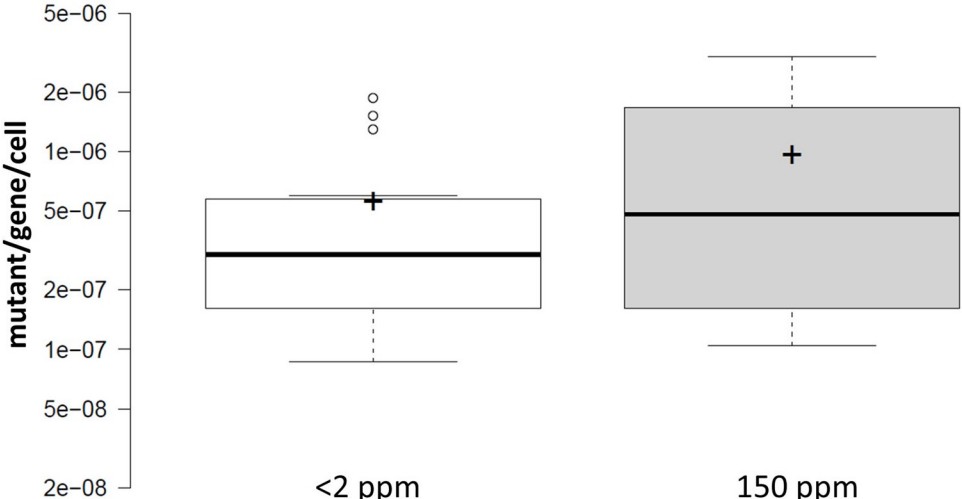

**Fig 5. Combined effect of D on the mutation frequencies measured in one of three genes (*cycA*, *ackA* or *galK*) of *E. coli*.** Center lines show the medians; box limits indicate the 25th and 75th percentiles as determined by R software; whiskers extend 1.5 times the interquartile range from the 25th and 75th percentiles, outliers are represented by dots; crosses represent sample means. The medians are not significantly different ($U = 94$, $z = -0.74$, $n_2 = n_{150} = 15$, P = .46, applying a Mann-Whitney U-test).

have collected examples of mutagenic and antimutagenic effects reported by other authors in various bacterial strains. Considering the effect of changes in the genotype, mutator mutants showed 5-100-fold increase in mutation rates measured in *E. coli* and *Caulobacter crescentus* using Rifampicin resistance (Rif$^R$)-based fluctuation assays [57]. In another work, frequencies of Rif$^R$, Spectinomycin resistant (Spc$^R$) or Nalidixic-acid resistant (Nal$^R$) mutants in pathogenic *E. coli* and *Salmonella* increased 10-500-fold due to the acquisition of mutator mutations [58]. The antimutator *dnaE915* allele reduced mutant frequencies 2.5-, 3.5- or 6.5-fold when using Rif$^R$, Nal$^R$ or *lacI* mutant detection systems, respectively. Chemical mutagens are very often characterized using the Ames test, which practically measures the frequency of various reverse mutants that have arisen in the presence of liver microsomes in a time frame of two days. Since these tests apply *Salmonella* cells grounded in stationary phase due to histidine-starvation, their results should not be directly compared to fluctuation tests typically conducted in exponentially-growing cultures. It is noteworthy however that in the seminal paper, the smallest change of mutant frequencies judged to be significant was a 4.4-fold increase [59]. In a study analyzing the mutagen-content of natural waters using a similar technique, the minimal effective dose required for *doubling* the mutant frequency (MED2) was declared as a relevant parameter to be calculated for all samples [60]. When reporting the mutagenicity of chemicals, many times the dose-dependent effect is demonstrated. For example, the mutagenicity of N4-aminocytidine measured in the 4–40 μM concentration range resulted in a 6-1000-fold increase in Rif$^R$ mutant frequencies [61]. To also mention examples of mutation-inhibition, certain fruit juices were classified as anti-mutagens based on their capability to reduce UV-induced mutant frequency 2–2.5-fold, measured using the Rif$^R$ assay [62]. What is common in all the papers cited above is that only those changes in mutation rate or mutation frequency are considered relevant which are greater than two-fold. Exceptions to this rule are rare, the only ones we found from other authors was a 1.8-fold and 1.5-fold increase in the frequency of Rif$^R$ mutants measured in Enterococci and coagulase-negative Staphylococci, respectively, isolated from patients that have gone through antibiotic treatment, compared to corresponding control strains [63]. Curiously however, repeating the Mann-Whitney test on the data manually extracted from their chart did not quite reconfirm the statistical significance of those two changes ($n_1$ = 11, $n_2$ = 18, $U$ = 195, $z$ = 1.91, P = .056 and $n_1$ = 14, $n_2$ = 20, $U$ = 125, $z$ = 1.15, P = 0.25, for Enterococci and coagulase-negative Staphylococci, respectively). Taking these into account, it is apparent that changes in the bacterial mutation rate or frequency smaller than two-fold are rarely, if ever reported to be significant in the literature today. We therefore conclude that according to current practice, natural D content of surface waters cannot be considered to have a mutagenic effect on bacteria. Perhaps future techniques developed to monitor mutation rates will provide data with smaller variation, which could allow a more stringent practice in defining the lower limits of mutagenicity.

There are two additional arguments to mention when considering our results. First, measuring mutational rates has a long history, and a large set of data is available in the literature. From reviews and summarizing datasets, it is clearly apparent that spontaneous mutation rates measured in various organisms are not limited to a small range, and especially not to a discrete value, but instead span six to seven orders of magnitude [64,65]. The variation of D-concentrations however is well within one order of magnitude in our biosphere, which suggests that D-abundance may influence, but is unlikely to "dictate" the rate of mutational events.

As a second factor however, we must note that our mutation rate assays deal with *E. coli* cells only, which may not allow full extrapolation of our results to eukaryotic cells or higher order organisms. Although the fundamental mechanisms of mutagenesis in *E. coli* are also present in higher order life forms, there are certainly differences in modifying factors, e.g. chromatin organization or DNA repair mechanisms. The average spontaneous mutation rate

of vertebrates is ca. 100-fold higher than that of prokaryotes [65], which could result in large differences in the relative contribution of certain physicochemical effects to the overall rate. The Ames test, despite being a good example of using bacteria to screen for chemicals mutagenic to humans, also calls our attention to the fact that many compounds require chemical transformation to become mutagenic, hence the need for liver microsomes [59]. Perhaps D may possess another type of indirect mutagenic property, although we are currently not aware of such information. We note nevertheless, that if a major difference in the effect of D on the mutation rate of bacteria and higher order organisms was revealed, it would also indicate the necessity to modify the hypothesis that globally and exclusively attributes single nucleotide variations to the presence of adjacent D-bonds within DNA base-pairs.

## Conclusions

Overall, we have carried out 24 pairs of fluctuation tests, utilizing a total of 680 bacterial cultures. Fifteen pairs were considered valid (corresponding to 460 cultures), but the extracted data was not sufficient to support the statistical significance of the antimutagenic effect of D-depletion. We nevertheless observed that changing the abundance of D from <2 ppm to 150 ppm has no detectable effect on the growth parameters of *E. coli* in glucose-minimal medium. In addition, although our observations are yet limited to bacteria, our data suggest that in general, the effect of D abundance on the mutation rate is unlikely to be deterministic.

## Supporting information

**S1 Fig. Effect of D on the mutation rate occurring in *E. coli* cultures.**
(PDF)

**S1 Table. PCR primers used in this study.**
(PDF)

**S2 Table. The mutation/culture values (*m*) and the total cell numbers (*N*) measured in the pairwise fluctuation analyses.**
(XLSX)

**S3 Table. Mutant frequencies (*f*) calculated from the valid tests of S2 Table.**
(XLSX)

**S1 File. Raw experimental data.**
(XLSX)

## Acknowledgments

We thank György Pósfai, Balázs Szalontai, Antal Kiss, Csaba Vizler and László G. Boros for helpful discussions and for proofreading various versions of the manuscript, as well as Csaba Tömböly and Attila Borics for technical help.

## Author Contributions

**Conceptualization:** Gábor Somlyai, Ildikó Somlyai.

**Funding acquisition:** Tamás Fehér.

**Investigation:** Walliyulahi Ajibola, Ildikó Karcagi.

**Methodology:** Walliyulahi Ajibola, Ildikó Karcagi, Tamás Fehér.

**Project administration:** Walliyulahi Ajibola, Ildikó Karcagi.

**Software:** Tamás Fehér.

**Supervision:** Tamás Fehér.

**Visualization:** Tamás Fehér.

**Writing – original draft:** Tamás Fehér.

**Writing – review & editing:** Gábor Somlyai, Ildikó Somlyai.

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
