## [Decision Letter · Decision Letter 0]

23 Dec 2020

PONE-D-20-36626

Deuterium-depletion has no significant impact on the mutation rate of Escherichia coli, invalidating the Double D-Bridge Hypothesis

PLOS ONE

Dear Dr. Tamás,

Thank you for submitting your manuscript to PLOS ONE. After careful consideration, we feel that it has merit but does not fully meet PLOS ONE’s publication criteria as it currently stands. Therefore, we invite you to submit a revised version of the manuscript that addresses the points raised during the review process.

We look forward to receiving your revised manuscript.

Kind regards,

Ashis K. Basu

Academic Editor

PLOS ONE

Additional Editor Comments:

This work can be published, pending revision. Specifically, avoid discounting the validity of the DDB hypothesis, as it is not supported by the data presented. Accordingly, please change the title of this article and the abstract. I recommend a careful rewriting of the manuscript. Please make sure to address all the comments of the reviewers.

"I have read the journal's policy and the authors of this manuscript have the following competing interests:

G.S. and I.S. are employed by HYD LLC.

HYD LLC for Cancer Research and Drug Development is involved in drug registration based on deuterium depletion and in the development of deuterium-depleted nutritional products.

The above company is not traded publicly nor does it distribute shares or annuities."

Reviewers' comments:

Reviewer's Responses to Questions

**Comments to the Author**

1. Is the manuscript technically sound, and do the data support the conclusions?

Reviewer #1: Partly

Reviewer #2: Yes

2. Has the statistical analysis been performed appropriately and rigorously? 

Reviewer #1: Yes

Reviewer #2: Yes

3. Have the authors made all data underlying the findings in their manuscript fully available?

Reviewer #1: Yes

Reviewer #2: Yes

4. Is the manuscript presented in an intelligible fashion and written in standard English?

Reviewer #1: Yes

Reviewer #2: Yes

5. Review Comments to the Author

Reviewer #1: a. MS presents a limited test of whether deuterium depleted water (ddw) and natural water systems will or will not yield the same spontaneous mutation rates (smr) for three e. coli strains

(1 generation, multiple copies of each).

b. they find very similar smrs for the three systems for both solvents and for the three stains for single generation growth curves.

c. the authors are concerned with ref 30 and the speculation that the natural abundance of D in earth water may be responsible for the smr. the current paper authors concoct the term "Double

D bridge hypothesis" to interpret the idea in ref 30 that if in double stranded (ds) dna, two adjacent hydrogen bonds have a D rather than an H, the prob of this happening can be estimated as

the square of the abundance of D in the natural water ..and this happens to be a number similar to the approximate smr...2 x 10^-8. So the term "Double D bridge hypothesis" is never stated in

Ref. 30. Also, the last sentence of the conclusions of the current paper says "In addition, we declare with relatively high confidence that the equality of the genomic mutation rate of humans and

C. Elegans with the theoretical frequency of A:T base pairs in the DNA connected by two D-bridges is only coincidental". Ref 30 says nothing about AT base pairs or D bridges.

d. since Ref. 30 was published 2006, which suggests the D in nat. abundance may contribute to the smr, (thru ds DNA, thru tunneling, through equilibrium effects) numerous studies have been

published on smrs. One of the most impressive is in PLOS biology (Krasovec, 2017) which presents a log log plot of smr vs population density (ml^-1) for all forms of biological life...the plot

covers 6 orders of magnitude in both directions. So smr is highly variable. Michael Lynch has shown the importance of studying the spectrum of spontaneous mutations through many generations

(thousands) (Lynch, Genetics 2015) and even in 2010 Trends in Genetics Lynch was pointing out that smrs were a 100 greater for vertebrates than bacteria.

e. Basically, the results on the 1 generation, bacterial systems, ddw vs nat water..look good--there appears to be little effect! What is not covered are the more rapidly mutating complex systems

over multiple generations.

f. I think it would be simple for the authors to remove "the Double D-bridge hypothesis" which is their characterization (something like DDW effect might be more appropriate)

and to point out clearly the limitations of their fine experiments (bacteria, one generation, etc)

Reviewer #2: Studies of the impact of D2O relative to H2O on biological systems has a long history that has been reviewed recently (Kselikova et al., Folia Microbiologica (2019) 64:673–681); this review cites evidence that unicellular organisms like E. coli tolerate a D2O environments (This article should be cited). However, it has been demonstrated that the mutagenic activity is enhanced upon incubation of E. coli cells in D2O. A Double D-Bridge mechanism was proposed to account for these observations; it is based on the hypothesis that the mutagenic activity is proportional to the deuteration of the two A:T hydrogen bonding atoms.

In this work, the mutation rates in three different E. coli genes were determined and interpreted within the framework of the DDB hypothesis. The experimental mutagenicity results and statistical analysis are rigorous and a useful contribution to the sparse literature on this topic.

However, I disagree with the conclusion that there is a disparity between the quantitative predictions of the DDB theory and the author’s experimental results. Their apparent and unstated assumption is that every polymerase-catalyzed insertion of the incorrect nucleotide is 100% mutagenic. Polymerase – catalyzed insertions of dNTP nucleotides opposite a template base is a competition between the four nucleotides; the dominant insertion is decided by the differences in equilibrium binding constants of the four dNTPs opposite the nucleotide in the template strand, before the nucleotidyl transfer step occurs. The canonical dNTP usually predominates and the correct nucleotide is inserted.

Deuteration may influence this equilibrium which means that a mutation-inducing event can occur with an efficiency much less than 100%. Since, this efficiency is unknown and can be much smaller than 100%, the DDB mechanism cannot be excluded. In other words, deuteration may not fully inhibit the error-free insertion of the canonical nucleotide which decreases the mutation rate.

In summary, I would recommend publication of this paper if this possibility is properly accounted for in their interpretation, and if they avoid discounting the validity of the DDB hypothesis which is not supported by the data presented (a change in the title and abstract is also called for). Finally, this work is important because it contradicts/corrects previous observations that reported > 1000-fold changes in mutagenicity attributed to deuterium effects which seem improbable.

6. PLOS authors have the option to publish the peer review history of their article (what does this mean?). If published, this will include your full peer review and any attached files.

Reviewer #1: No

Reviewer #2: No

---

## [Author Response · Author response to Decision Letter 0]

9 Feb 2021

Response to Reviewers

Reviewer 1.

a. MS presents a limited test of whether deuterium depleted water (ddw) and natural water systems will or will not yield the same spontaneous mutation rates (smr) for three e. coli strains

(1 generation, multiple copies of each).

Response: 

We kindly call the reviewer’s attention to the fact that in all three genetic tests, the E. coli strains grew not for one generation, but for 13-14 generations. (Starting cell density: ~10^5/ml, final cell density: ~10^9/ml; ratio=10,000; log(2)10,000=13.28) We added the following sentence to the Methods to clarify this issue to the readers: ”Considering that the final densities were 10^9 cells/mL, this growth corresponded to ca. 13 cell duplications or generations.”

b. they find very similar smrs for the three systems for both solvents and for the three stains for single generation growth curves.

c. the authors are concerned with ref 30 and the speculation that the natural abundance of D in earth water may be responsible for the smr. the current paper authors concoct the term "Double

D bridge hypothesis" to interpret the idea in ref 30 that if in double stranded (ds) dna, two adjacent hydrogen bonds have a D rather than an H, the prob of this happening can be estimated as

the square of the abundance of D in the natural water ..and this happens to be a number similar to the approximate smr...2 x 10^-8. So the term "Double D bridge hypothesis" is never stated in

Ref. 30. Also, the last sentence of the conclusions of the current paper says "In addition, we declare with relatively high confidence that the equality of the genomic mutation rate of humans and

C. Elegans with the theoretical frequency of A:T base pairs in the DNA connected by two D-bridges is only coincidental". Ref 30 says nothing about AT base pairs or D bridges.

Response: 

Concerning “A:T base pairs”, we accept the criticism and delete this phrase. 

Concerning the name of the hypothesis: Pedersen et al. did not give a name to their hypothesis. For easy and clear referral and to preserve the fluency of the text, we decided to call it "Double D bridge hypothesis". We think this was not misleading, for in Line 82 of the first version, we linked this term to the reference. But since this is not an official name, nor is it widely used, it is probably improper to use it in our title and Abstract. We therefore accept the criticism and modified the title, and removed this phrase from the Abstract and the main text. 

Concerning “D-bridge”: Pedersen et al. use a more sophisticated phrasing, e.g. “...base-pairing hydrogen bonds have D rather than H in the H-bonds...”. This also required a simple and fluent replacement. Since H-bonds connecting two nucleobases are often called hydrogen-bridges in chemistry, replacing H with D would be called deuterium-bridges, or D-bridges. We assume this is understood by all readers who have read about H-bridges. We nevertheless added an explanation to make this clear to all readers: “The basis of this explanation is the fact that the probability of base pairs harboring two adjacent deuterium-bonds (D-bridges) instead of hydrogen-bonds (H-bridges) (i.e. 1.5 x 10-4 squared) is approximately equal to the mutation rate (~2 x 10-8 mutations/nucleotide/generation) calculated for humans and Caenorhabditis elegans [31]”. The last sentence of the Conclusion has been rephrased.

d. since Ref. 30 was published 2006, which suggests the D in nat. abundance may contribute to the smr, (thru ds DNA, thru tunneling, through equilibrium effects) numerous studies have been

published on smrs. One of the most impressive is in PLOS biology (Krasovec, 2017) which presents a log log plot of smr vs population density (ml^-1) for all forms of biological life...the plot

covers 6 orders of magnitude in both directions. So smr is highly variable. Michael Lynch has shown the importance of studying the spectrum of spontaneous mutations through many generations

(thousands) (Lynch, Genetics 2015) and even in 2010 Trends in Genetics Lynch was pointing out that smrs were a 100 greater for vertebrates than bacteria.

Response:

We express our sincere gratitude to the Reviewer for mentioning both papers. Reviews written on spontaneous mutation rates, or the log plot of Krasovec et al. 2017, are a good indication that smrs span several orders of magnitude. This is a strong argument against the model of Pedersen et al., who argue that the D/H ratio is deterministic for the smr of any organism. It is easy to find smrs on that plot which equal the theoretical chances of having two D-bonds within a base-pair, but it is visible that smrs can also be 100-fold lower or 10,000-fold higher. We included this line of thought as the penultimate paragraph of the Discussion in the revised manuscript.

e. Basically, the results on the 1 generation, bacterial systems, ddw vs nat water..look good--there appears to be little effect! What is not covered are the more rapidly mutating complex systems

over multiple generations.

f. I think it would be simple for the authors to remove "the Double D-bridge hypothesis" which is their characterization (something like DDW effect might be more appropriate)

and to point out clearly the limitations of their fine experiments (bacteria, one generation, etc)

Response:

We emphasize in the ultimate paragraph of the revised Discussion that our experimental system has its limitations, necessitating tests in higher-order organisms. We repeat that our system utilizes 13 generations of bacterial growth. 

In our opinion, DDW effect (i.e. the effect of growth in D-depleted medium on mutation rates) irrespective of whether it’s measurable or negligible, is not the same as the hypothesis explaining the potential effect (irrespective whether we call it "the Double D-bridge hypothesis" or “Pedersen’s hypothesis”). Therefore, we believe that the recommended change would mean using a phrase in the wrong context. The term “Double D-bridge hypothesis" is nevertheless avoided in the revised manuscript. 

Reviewer #2: Studies of the impact of D2O relative to H2O on biological systems has a long history that has been reviewed recently (Kselikova et al., Folia Microbiologica (2019) 64:673–681); this review cites evidence that unicellular organisms like E. coli tolerate a D2O environments (This article should be cited). 

Response:

We highly appreciate calling our attention to this review article. We included it as a new reference in the Introduction.

However, it has been demonstrated that the mutagenic activity is enhanced upon incubation of E. coli cells in D2O. A Double D-Bridge mechanism was proposed to account for these observations; it is based on the hypothesis that the mutagenic activity is proportional to the deuteration of the two A:T hydrogen bonding atoms.

In this work, the mutation rates in three different E. coli genes were determined and interpreted within the framework of the DDB hypothesis. The experimental mutagenicity results and statistical analysis are rigorous and a useful contribution to the sparse literature on this topic.

However, I disagree with the conclusion that there is a disparity between the quantitative predictions of the DDB theory and the author’s experimental results. Their apparent and unstated assumption is that every polymerase-catalyzed insertion of the incorrect nucleotide is 100% mutagenic. 

Polymerase – catalyzed insertions of dNTP nucleotides opposite a template base is a competition between the four nucleotides; the dominant insertion is decided by the differences in equilibrium binding constants of the four dNTPs opposite the nucleotide in the template strand, before the nucleotidyl transfer step occurs. The canonical dNTP usually predominates and the correct nucleotide is inserted.

Deuteration may influence this equilibrium which means that a mutation-inducing event can occur with an efficiency much less than 100%. Since, this efficiency is unknown and can be much smaller than 100%, the DDB mechanism cannot be excluded. In other words, deuteration may not fully inhibit the error-free insertion of the canonical nucleotide which decreases the mutation rate.

Response:

We agree with the reviewer’s interpretation of Pedersen et al’s paper, concerning that they assume that every polymerase-catalyzed insertion of the incorrect nucleotide is 100% mutagenic. Furthermore, they assume that if two hydrogens of the nucleotide responsible for H-bond formation are actually deuterons (theoretical chance: 2E-8), the rate of its misincorporation is ~100%. If it was far less than 100%, the equality of the above number with the mutation rates observed in two systems would be coincidental. We modified the text to emphasize that we do not exclude the role of double D-bond formation in the misincorporation of nucleotides (lines 405-412 of the revised Discussion with markup). We however believe that Pedersen meant a 100% determination of mutation rates by the double D-bond abundance (which is determined by D-abundance):

i). In the first sentence of their Abstract, they write:

“...the spontaneous mutation rate, the core of evolution theory, may be dictated by the deuterium/

hydrogen (D/H) abundance ratio.” 

ii) last sentence of their section 3: “It is not difficult to imagine that along the dsDNA, there will be a statistical probability of 2x10^-8 that two D’s will locate in adjacent H-bonds, each contributing about 2 kcal/mol to the stability of the DNA. The magnitude of the energy lowerings for D-containing DNA base pairs is significant and certainly large enough to turn on or off a key reactive process.” This binary on/off switch describes a deterministic model of point mutagenesis, determined by the double D-bond abundance (determined by the D/H ratio). We maintain our opinion that this deterministic model does not stand, according to our experimental data.

In summary, I would recommend publication of this paper if this possibility is properly accounted for in their interpretation, and if they avoid discounting the validity of the DDB hypothesis which is not supported by the data presented (a change in the title and abstract is also called for). Finally, this work is important because it contradicts/corrects previous observations that reported > 1000-fold changes in mutagenicity attributed to deuterium effects which seem improbable.

Response:

By all means, we do not intend to discount the work of Pedersen et al. We highly value their work, a publication which we learned a lot from. In the last sentence of their paper however, they encourage others to test their hypothesis. We did that and our data does not support a deterministic model. We indicate this, but nevertheless included in the revised manuscript the possibility of nucleotide-misincorporations being affected (but not dictated) by D-abundance. We also modified the title (also to fulfill the expectations of the other reviewer), which now makes it clear that our data contradict with the deterministic nature of their model, not their full model. We explain in the revised text, that deuteration can affect both the kinetics and the equilibrium energy states of nucleotide incorporation, thereby potentially increasing the chances of misincopropration. We also modified the Abstract to be in line with this.

Finally, we thank both reviewers for their constructive criticism and valuable recommendations. We enjoyed the scientific discussion, and strongly believe that the reviewers’ work significantly improved our manuscript.

---

## [Decision Letter · Decision Letter 1]

22 Feb 2021

Deuterium-depletion has no significant impact on the mutation rate of Escherichia coli, deuterium abundance therefore has a probabilistic, not deterministic effect on spontaneous mutagenesis

PONE-D-20-36626R1

Dear Dr. Tamás,

We’re pleased to inform you that your manuscript has been judged scientifically suitable for publication and will be formally accepted for publication once it meets all outstanding technical requirements.

Kind regards,

Ashis K. Basu

Academic Editor

PLOS ONE

Additional Editor Comments (optional):

Reviewers' comments:

Reviewer's Responses to Questions

**Comments to the Author**

1. If the authors have adequately addressed your comments raised in a previous round of review and you feel that this manuscript is now acceptable for publication, you may indicate that here to bypass the “Comments to the Author” section, enter your conflict of interest statement in the “Confidential to Editor” section, and submit your "Accept" recommendation.

Reviewer #1: All comments have been addressed

Reviewer #2: All comments have been addressed

2. Is the manuscript technically sound, and do the data support the conclusions?

Reviewer #1: Yes

Reviewer #2: Yes

3. Has the statistical analysis been performed appropriately and rigorously? 

Reviewer #1: I Don't Know

Reviewer #2: Yes

4. Have the authors made all data underlying the findings in their manuscript fully available?

Reviewer #1: Yes

Reviewer #2: Yes

5. Is the manuscript presented in an intelligible fashion and written in standard English?

Reviewer #1: Yes

Reviewer #2: Yes

6. Review Comments to the Author

Reviewer #1: Accept. Authors did a nice job of answering questions. Accept. Authors did a nice job of answering questions.

Accept. Authors did a nice job of answering questions. Accept. Authors did a nice job of answering questions.

Reviewer #2: The author's response to my comments are excellent and fully satisfactory.

7. PLOS authors have the option to publish the peer review history of their article (what does this mean?). If published, this will include your full peer review and any attached files.

Reviewer #1: No

Reviewer #2: No

---

## [Editor Report · Acceptance letter]

26 Feb 2021

PONE-D-20-36626R1 

Deuterium-depletion has no significant impact on the mutation rate of *Escherichia coli*, deuterium abundance therefore has a probabilistic, not deterministic effect on spontaneous mutagenesis 

Dear Dr. Fehér:

I'm pleased to inform you that your manuscript has been deemed suitable for publication in PLOS ONE. Congratulations! Your manuscript is now with our production department. 

Kind regards, 

on behalf of

Dr. Ashis K. Basu 

Academic Editor

PLOS ONE